# Phosphatidylethanolamines Are Associated with Nonalcoholic Fatty Liver Disease (NAFLD) in Obese Adults and Induce Liver Cell Metabolic Perturbations and Hepatic Stellate Cell Activation

**DOI:** 10.3390/ijms24021034

**Published:** 2023-01-05

**Authors:** Samaa Shama, Hyejeong Jang, Xiaokun Wang, Yang Zhang, Nancy Nabil Shahin, Tarek Kamal Motawi, Seongho Kim, Samer Gawrieh, Wanqing Liu

**Affiliations:** 1Department of Pharmaceutical Sciences, Eugene Applebaum College of Pharmacy and Health Sciences, Wayne State University, Detroit, MI 48201, USA; 2Cell-Based Analysis Unit, Reference Laboratory, Egyptian Drug Authority, Cairo 12618, Egypt; 3Biostatistics and Bioinformatics Core, Department of Oncology, Karmanos Cancer Institute, Wayne State University, Detroit, MI 48201, USA; 4Department of Oncology, Wayne State University School of Medicine, Detroit, MI 48201, USA; 5Department of Biochemistry, Faculty of Pharmacy, Cairo University, Cairo 11562, Egypt; 6Division of Gastroenterology and Hepatology, Department of Medicine, Indiana University School of Medicine, Indianapolis, IN 46202, USA; 7Department of Pharmacology, Wayne State University School of Medicine, Detroit, MI 48201, USA

**Keywords:** non-alcoholic fatty liver disease, obesity, phospholipids, phosphatidylethanolamine, mitochondria, fibrosis, cell migration

## Abstract

Pathogenesis roles of phospholipids (PLs) in nonalcoholic fatty liver disease (NAFLD) remain incompletely understood. This study investigated the role of PLs in the progression of NAFLD among obese individuals via studying the alterations in serum PL composition throughout the spectrum of disease progression and evaluating the effects of specific phosphatidylethanolamines (PEs) on FLD development in vitro. A total of 203 obese subjects, who were undergoing bariatric surgery, were included in this study. They were histologically classified into 80 controls (C) with normal liver histology, 93 patients with simple hepatic steatosis (SS), 16 with borderline nonalcoholic steatohepatitis (B-NASH) and 14 with progressive NASH (NASH). Serum PLs were profiled by automated electrospray ionization tandem mass spectrometry (ESI-MS/MS). HepG2 (hepatoma cells) and LX2 (immortalized hepatic stellate cells or HSCs) were used to explore the roles of PL in NAFLD/NASH development. Several PLs and their relative ratios were significantly associated with NAFLD progression, especially those involving PE. Incubation of HepG2 cells with two phosphatidylethanolamines (PEs), PE (34:1) and PE (36:2), resulted in significant inhibition of cell proliferation, reduction of mitochondrial mass and membrane potential, induction of lipid accumulation and mitochondrial ROS production. Meanwhile, treatment of LX2 cells with both PEs markedly increased cell activation and migration. These effects were associated with a significant change in the expression levels of genes involved in lipogenesis, lipid oxidation, autophagy, apoptosis, inflammation, and fibrosis. Thus, our study demonstrated that elevated level of PEs increases susceptibility to the disease progression of obesity associated NAFLD, likely through a causal cascade of impacts on the function of different liver cells.

## 1. Introduction

Non-alcoholic fatty liver disease (NAFLD) is the most common chronic liver disease with a global prevalence of about 25% [1]. NAFLD comprises a spectrum of diseases ranging from lipid accumulation in the liver (steatosis) to lipid accumulation combined with inflammation, fibrosis, and cell damage (non-alcoholic steatohepatitis, NASH), which may progress to liver cirrhosis and even hepatocellular carcinoma [2]. Similarly, to other complex diseases, the pathogenesis and history of NAFLD appears to be determined by an intricate interplay between genes, gene products, and environmental signals [3,4]. NAFLD and obesity are highly correlated, though NAFLD is also prevalent in lean individuals. NAFLD in obese individuals is newly defined as metabolic associated fatty liver disease (MAFLD) which may have mechanisms distinct from those in lean people [5,6].

Phospholipids (PLs), including phosphatidylcholine (PC), phosphatidylethanolamine (PE), phosphatidylserine (PS), sphingomyelin (SM), etc., are chief lipid constituents in animal cell membranes, serving as a matrix that supports numerous membrane proteins [7]. Apart from their role in bilayer formation, PLs are also considered as signaling molecules for external stimuli response [8], regulating exocytosis, chemotaxis, and cytokinesis, surrounding particles to form phagosomes in phagocytosis as well as participating in the generation of vacuoles in endocytosis [9]. Additionally, PLs are involved in the regulation of processes related to growth, synaptic transmission, and immune monitoring [10,11,12]. Another function of PLs is the assembling of circulating lipoproteins essential for transport of triglycerides and cholesterol in blood [13]. Moreover, in the gallbladder, they emulsify cholesterol with bile acid forming micelles facilitating the absorption of fatty substances [14]. They act as smoothing agents for joints, alveoli, and other body parts [15]. Some PLs such as PE and cardiolipin are involved in the process of lipid transportation between the endoplasmic reticulum and the mitochondria [12]. PE and cardiolipin can induce stress and promote negative membrane curvature on the inner mitochondrial membrane, allowing the assembly of a membrane complex protein which consequently helps in fusion/fission of the mitochondrial membrane [16]. Therefore, PLs crucially maintain lipid homeostasis either in the mitochondrial membrane or in other cellular membranes [17].

An alteration in hepatic lipid metabolism is considered a hallmark of NAFLD. Remodeling in the composition of PLs in the blood or liver has been associated with liver injuries. Previous studies have collectively demonstrated the association between perturbation in PE levels and NAFLD, with a decreased PC/PE ratio associated with both liver steatosis and NASH [18,19], while increased level of PE was specifically associated with NASH [19] and disease progression [20]. This is, at least in part, attributed to the insufficiency of the phosphatidylethanolamine N-methyltransferase (PEMT), a key enzyme converting PE to PC [21,22,23,24]. Increased plasma level of PE was also observed in children with hepatosteaosis [25]. However, how PLs play their role in obesity-associated NAFLD remains less studied. More importantly, the causal role of PE in the pathogenesis of NAFLD was not fully explored. Our unique study investigated the importance of PLs, especially PE, in the pathogenesis of NAFLD.

The aim of the present study was to identify key PLs associated with the progression of obesity-associated NAFLD through lipidomic analysis of collected serum samples. Following this, the causal role of two PEs in the induction of cellular function changes underlying the increased risk for liver injuries was explored using cell models in vitro.

## 2. Results

### 2.1. Demographic, Histological, and Biochemical Features of the Study Subjects

The demographic characteristics, distribution of histological features and biochemical findings of the study subjects are summarized in Table 1, Table 2 and Table 3, respectively. There were no significant differences in age, sex, race, and body weight among groups. We observed that 34% (*p* = 0.008), 50% (*p* = 0.008), and 64% (*p* ˂ 0.001) of SS, B-NASH and NASH patients, respectively, were diabetic as compared to the control. Meanwhile, 79% of NASH patients were hypertensive relative to the control.

### 2.2. Elevation of Serum PLs in SS, B-NASH, and NASH

A total of 203 species of PLs and sphingomyelin (SM) were identified and quantified in serum by using ESI-MS/MS. In our study, we investigated a relation between the levels of several PL species and NAFLD. A significant increase was observed in PE and cholesterol ester (CE) species including PE (34:1), (36:2), (34:2) and C16:1 CE in NASH, B-NASH and SS compared to control subjects (Figure 1A–D). The ratio of lysophosphatidylcholine (LPC)/PEs and LPC/C16:1 CE showed a dramatic reduction between NAFLD and control groups (Figure 1E–I).

### 2.3. PEs Inhibited Cell Viability in HepG2 Cells

We set out to examine the causal role of two PEs on liver cell functions. While our lipidomic analysis does not generate information about the acyl composition of the PEs, we chose 1-palmitoyl-2-oleoyl-sn-glycerol phosphoethanolamine (34:1) and 1,2-dioleoyl-sn-glycero-3 phosphoethanolamine (36:2) since these two PEs are the most abundant PE (34:1) and PE (36:2) based on previous studies [27,28]. The effect of the two PEs on cell viability was performed by the CCK-8 assay. Both PEs significantly decreased cell viability in a dose- and time-dependent manner. HepG2 growth was markedly decreased by 14.2%, 27.3% and 41.7%, compared to the control after incubation with 0.25 mM, 0.5 mM, and 1 mM PE (34:1), respectively, after 48 h of treatment (Figure 2A). Similarly, after treatment with 0.25 mM, 0.5 mM, and 1 mM PE (36:2) for 48 h, the cell viability was decreased by 13.8%, 28.3% and 37.4%, respectively compared to the control (Figure 2A). Over different incubation periods (24, 48 and 72 h) of 1 mM PE (34:1) and PE (36:2), the cells showed a significant inhibition in cell growth after 48 and 72 h, compared to the control group (Figure 2B).

### 2.4. PEs Increased Lipid Droplet Accumulation in HepG2 Cells

The intracellular content of neutral lipids such as cholesterol esters and triglycerides that are packaged in lipid droplets (LDs) was estimated by fluorescence intensity and fluorescence microscopy after BODIPY staining of the cytoplasmic LDs. Treatment of HepG2 cells with 1 mM PE for 48 h significantly increased the staining of LDs compared to the control as observed by fluorescence microscopy. A highly significant difference in fluorescence intensity (*p* < 0.001) was detected in both PE (34:1) and PE (36:2) treated cell cultures as compared to the control group (Figure 3).

### 2.5. PEs Increased Mitochondrial Dysfunction in HepG2 Cells

To ascertain whether there are alterations in mitochondrial functions in response to PE treatment in HepG2 cells, we evaluated mitochondrial mass, membrane potential as well as mitochondrial reactive oxygen species (ROS) production. Mitochondrial mass and membrane potential were measured by using MitoTracker green and MitoTracker red, respectively. A significant reduction in both mitochondrial mass and membrane potential was observed under the fluorescence microscope after treatment with 1 mM PE for 48 h (Figure 4A). A reduction in fluorescence intensity of mitochondrial mass was detected in PE (34:1) and PE (36:2) treated cell cultures (*p* ˂ 0.001) (Figure 4B). Additionally, mitochondrial membrane potential was also significantly reduced in HepG2 cells treated with PE (34:1) and PE (36:2) as measured by fluorescence intensity (*p* ˂ 0.001) (Figure 4C). Mitochondrial reactive oxygen was measured by MitoSox red as the mitochondrial superoxide indicator. Mitochondria in both PEs-treated cells produced more superoxide than control cells as evidenced by more than two times higher fluorescence intensity compared to the control [(*p* ˂ 0.001 for PE (34:1) & *p* ˂ 0.01 for PE (36:2)] (Figure 5).

### 2.6. PE Altered Lipid Homeostasis Gene Expression in HepG2 Cells

To elucidate the possible mechanism mediating the observed effect of PEs, we next examined the expressions of genes involved in hepatic lipid homeostasis in cells treated with 1 mM PE for 48 h compared to the control group. Quantitative RT-PCR (qRT-PCR) data from PE (34:1) and PE (36:2) treated cells revealed significantly increased expression of carnitine palmitoyltransferase (*CPT*) (~1.5-fold, *p* < 0.05 for both PEs), peroxisome proliferator-activated receptor-alpha (*PPARa*) (~1.7-fold, *p* < 0.001 and ~1.5-fold, *p* ˂ 0.01 for PE (34:1) and PE (36:2), respectively), diglyceride acyltransferase (*DGAT1*) (~2-fold, *p* < 0.01 for both PEs), fatty acid synthase (*FASN*) (~2-fold, *p* < 0.001 and ~1.8-fold, *p* < 0.001 for PE (34:1) and PE (36:2), respectively) and sterol regulatory element-binding protein (*SREBP*) (~1.5-fold, *p* < 0.01 and *p* < 0.05 for PE (34:1) and PE (36:2), respectively) compared to the control. On the contrary, the results revealed a significant downregulation in the expression of farnesoid X receptor gene (*FXR*) in groups treated with PE (34:1) and (36:2) (~0.7-fold, *p* < 0.01) (Figure 6).

### 2.7. PEs Altered the Expression of Mitophagy and Apoptosis/Cell Proliferation-Related Genes in HepG2 Cells

To determine whether these PEs have an impact on mitophagy and apoptosis pathways, we examined the mRNA levels of genes involved in both processes after treatment of HepG2 cells with 1 mM PE for 48 h. The results revealed significantly increased gene expression levels of microtubule-associated protein (*LC3*) [~1.7-fold, *p* < 0.01 and ~1.5-fold, *p* < 0.05 for PE (34:1) and PE (36:2), respectively], BCL2-associated X protein (*BAX*) [~2.4-fold, *p* < 0.01 and ~1.7-fold, *p* < 0.05 for PE (34:1) and PE (36:2), respectively], p21-activated kinase 2 (*PAK2*) (~1.6-fold, *p* < 0.001 for both PEs) and cytochrome C (*CYCS*) [1.6-fold, *p* < 0.01 and *p* < 0.05 for PE (34:1) and PE (36:2), respectively] compared to the control group. Conversely, the expression of B-cell lymphoma 2 (*BCL2*) was significantly downregulated (~0.5-fold, *p* < 0.01 for both PEs) after treatment of HepG2 cells with 1 mM PE for 48 h. Similarly, a remarkable reduction in the relative gene expression of mechanistic target of rapamycin kinase (*mTOR*) [~0.7-fold, *p* < 0.001 and ~0.9-fold, *p* < 0.001 for PE (34:1) and PE (36:2), respectively] was observed in cells treated with PE (34:1) and PE (36:2). However, the expression of carbohydrate response element binding protein (*ChREBP*) was not significantly altered in response to PE treatment compared to the control group (Figure 6).

### 2.8. PEs Induced LX2 Cell Migration

The hepatic stellate cell (HSC) is the key cell type responsible for fibrosis in acute and chronic liver injuries. To examine the activation of LX2 by PE, we performed the cell migration assay using transwell by incubating cells with 1 mM PE for 24 h. The assay showed dramatic increments in the number of stained migrated cells after treatment with PE (34:1) and PE (36:2) (Figure 7A). The data showed a significant ~2-fold elevation in absorbance after incubation with PE (34:1) and PE (36:2) (*p* < 0.001 and *p* ˂ 0.01, respectively) (Figure 7B).

### 2.9. PE Treatments Increased mRNA Expression of Inflammation and Fibrosis-Related Genes in LX2 Cells

The activation of LX2 detected in the present study by the cell migration assay after incubation with PE indicated the induction of inflammation and fibrosis. To validate these actions, we examined the effect of PE on the expression of genes involved in both processes. The results showed overexpression of fibrotic marker genes including α-smooth muscle actin (*α-SMA*) [~2.9-fold, *p* < 0.001 and ~2.4-fold, *p* < 0.001 for PE (34:1) and PE (36:2), respectively], collagen type I alpha 1 (*COL1A1*) [~1.2-fold, *p* < 0.05 and ~1.3-fold, *p* < 0.05 for PE (34:1) and PE (36:2), respectively], collagen type III alpha 1 (*COL3A1*) [~1.9-fold, *p* < 0.001 and~1.7-fold, *p* < 0.01 for PE (34:1) and PE (36:2), respectively], TIMP metallopeptidase inhibitor 1 (*TIMP1*) [~1.9-fold, *p* < 0.01 and ~1.7-fold, *p* < 0.01 for PE (34:1) and PE (36:2), respectively), and transforming growth factor beta-1 (*TGF-beta*) (~1.3-fold, *p* < 0.01 for both PEs). Correspondingly, PE enhanced the expression of inflammatory markers including tumor necrosis factor-alpha (*TNF-α*) [~1.4-fold, *p* ˂ 0.001 and ~1.6-fold, *p* < 0.001 for PE (34:1) and PE (36:2), respectively]. On the other hand, the mRNA level of TIMP metallopeptidase inhibitor 3 (*TIMP3*) remained unchanged and interleukin-6 (*IL6*) was upregulated only after PE (34:1) treatment (~1.2-fold, *p* ˂ 0.05) as compared to the control group (Figure 8).

## 3. Discussion

Phospholipids (PLs) play fundamental roles in numerous biochemical reactions and different metabolic pathways. PL composition alterations in both blood and liver tissue of NAFLD patients as compared to healthy individuals have been observed previously [15,29]. However, it remains less investigated how the blood PL composition might be associated with NAFLD progression among obese individuals. More importantly, the actual role of PLs, especially PEs, in NAFLD as to whether they are considered as a causal factor, or just a consequence of the disease remains incompletely studied. In the current study, we investigated the role of PLs in obesity-associated NAFLD progression in vivo and explored the mechanism underlying this role in vitro.

Based on our results, a remarkable change was observed in serum PLs that varied significantly between SS, B-NASH and NASH subjects in obese individuals. The most dramatically changed PL species and their ratios observed in our study were CE, PE, LPC/PE, and LPC/CE. As demonstrated in Figure 1, increased levels of three specific PEs and CE while decreased ratios of LPC/PE were associated with the disease progression. Our data showed a distinguishable increase in PE species containing saturated fatty acid (SFA) and monounsaturated fatty acid (MUFA) in NAFLD subjects compared to control. The elevation of MUFA/SFA containing PLs may be due to increased insulin secretion as well as high dietary intake in obese subjects [30] which consequently activates de novo lipogenesis and accumulation of MUFA/SFA PLs fraction in blood [25,31].

Lysophosphatidylcholines (LPCs) are lipids derived from PCs mediated by phospholipase A2 (PLA2) which cleaves PC to produce LPC and free fatty acid, and/or by lecithin-cholesterol acyltransferase (LCAT) which transfers fatty acids from PC to free cholesterol to produce LPC and CE. LPC can also be converted back to PC mediated by lysophosphatidylcholine acyltransferase (LPCAT) in the presence of acyl-CoA [32]. Therefore, our findings may suggest an imbalanced activity of these enzymes in obesity-associated NAFLD development. Indeed, altered function of both LCAT and LPCAT3 have been found to be associated with NAFLD [33,34]. Unlike previous studies, we did not observe a significant association between PC/PE ratio and NAFLD disease severity in our cohort, which may be related to the obese background of our cohort, since PC/PE ratio seemed to be more likely associated with NAFLD in lean individuals [20,21], though animal studies might not support this notion [35]. Further studies are needed to validate and clarify the role of PC, PE, and LPC in NAFLD in both lean and obese individuals.

Our data also suggest that the LPC/CE and LPC/PE ratios that are significantly associated with NAFLD disease progression in our samples may be primarily driven by the increased CE and PE levels, given that CE or PE alone was associated with NAFLD, while LPC was not. This suggests that accumulated CE and PE are likely more causal to the disease progression. Compared to what have been broadly studied on the role of cholesterol homeostasis in NAFLD and NASH, the causal role of PE in the development and progression of NASH remains largely elusive. We thus set out to investigate the impact of PE on underlying features of NASH histology, i.e., steatosis, cell proliferation, and hepatic stellate cell (HSC) activation in vitro.

Since HepG2 and LX2 have been widely used to model NAFLD in vitro [36], we utilized these two cell models in our study to elucidate the effect of PE on the development of the disease. In the current study, HepG2 cells were used to study lipid accumulation, apoptosis, and mitochondrial dysfunction, while LX2 cells were used to assess inflammation and fibrosis [37]. PEs containing SFA/MUFA side chains with 16 and 18 carbons are the most abundant lipid molecular species in the human liver [27]. Moreover, previous investigators showed that PE species play a basic role in the pathology of several diseases [38]. Oleic acid (OA) and palmitic acid (PA) are the most abundant MUFA and SFA in human tissues and are commonly used in vitro and in vivo to stimulate steatosis [28,39,40]. Based on our lipidomic data, the PEs that mostly showed significant changes in NASH were PE species containing SFA and MUFA. Accordingly, we performed the in vitro study with PE (34:1) and (36:2) that contain OA (C18 MUFA) and PA (C16 SFA) moieties as examples. Treating HepG2 cells with these two representative PEs significantly decreased cell viability but increased intracellular neutrolipid accumulation coupled with increased impairments in mitochondrial function. The mitochondrial dysfunction was characterized by reduced mitochondrial mass and their membrane potential. This dysfunction was also associated with increased ROS production. Previous studies using HepG2 and other cells also consistently demonstrated that PE induces cell apoptosis and mitochondrial dysfunction. However, these studies did not specify the acyl composition of PEs used to treat the cells [41,42]. Our study confirmed these previous findings with two specific PEs, and further demonstrated that HepG2 cells treated with these two PEs also developed increased neutrolipid accumulation and ROS production. In correspondence with these results, our data showed significantly altered expression of genes involved in mitophagy (*LC3*, and *mTOR*), apoptosis/cell proliferation (*BAX*, *BCL2*, *PAK2*, *CYCS*, and *ChREBP*), beta-oxidation (*CPT* and *PPARα*) and lipogenesis (*DGAT*, *FASN*, *SREBP* and *FXR*). Again, these gene expression patterns also confirmed the previously observed impairment in apoptosis/cell proliferation associated with PE treatment [41,42], while the increased expression of lipogenesis genes following PE treatments may combine with the impaired mitochondrial function and together lead to neutrolipid accumulation and ROS production. These cellular impacts of the two PEs well match the pathogenic features, i.e., cell death, oxidative stress, lipid accumulation, and mitochondrial dysfunction that are underlying NAFLD [43].

Hepatic stellate cells (HSC) activation plays a central role in hepatic fibrosis during the development of NASH. Upon activation, they differentiate into proliferative and migratory myofibroblasts that accumulate in areas of liver injury producing extracellular matrix (ECM) components and cytokines [44]. The data presented herein for the cell migration assay revealed that incubation of LX2 cells with both PEs resulted in significantly increased cell stimulation and subsequently migration. In parallel, gene expression analysis indicated a relative upregulation of genes involved in fibrosis (*α-SMA*, *Col3A*, *Col1A*, *TIMP1*, *TIMP3* and *TGF-beta*) and inflammation (*TNFα* and *IL6*).

These impacts of PE again match the pathogenic changes underlying NASH. These observations, coupled with the positive correlation between these PEs and the progression of NAFLD, highlighted that elevated PEs are causal factors involved in the development of NASH. The consistent results between these two PEs suggested that SFA and MUFA in PEs may exert a similar impact on liver injury and disease progression of NAFLD.

Our study has a few limitations. First, our study does not have an independent obese NAFLD sample set to further confirm our findings. Therefore, our observations may only be limited to our sample set, and further replications would help to validate our results. Second, we only explored the potential causal role of two common PEs containing SFA and MUFA in vitro. HepG2 and LX2 may not be ideal in vitro models for NAFLD and NASH as well. It is unclear whether other PEs, e.g., those with very long chain polyunsaturated fatty acid (PUFA) moieties play different roles in the development of NASH, especially in vivo. Third, further characterizing the protein level of marker genes in the model may provide additional information for the impact of PEs. Fourth, our observations at the causal consequences of PEs on liver cell injuries and HSC activation should be further validated by inhibiting the action of PEs. Unfortunately, key mediators underlying the biological impact of PE signaling are currently unknown. Future studies should further examine the impact of different PE species in animal models and critical signaling mediators for the function of PEs. Nevertheless, our study provides novel insights into the mechanism of action of PE in NAFLD development and generated new hypotheses warranting further investigations.

In conclusion, the current study revealed that elevated levels of PEs containing SFA and MUFA are associated with the development of NAFLD and NASH, while these PEs are highly likely involved in the development of liver damage and disease progression processes by inducing lipid accumulation, mitochondrial dysfunction, cell growth inhibition, oxidative stress as well as activation of HSCs.

## 4. Materials and Methods

### 4.1. Materials

Chemicals and cell culture reagents (Dulbecco’s Modified Eagle’s medium, penicillin/streptomycin, 1-palmitoyl-2-oleoyl-sn-glycerol phosphoethanolamine (34:1), 1,2-dioleoyl-sn-glycero-3 phosphoethanolamine (36:2) and fetal bovine serum (FBS)) were obtained from Sigma (St. Louis, MO, USA) and Gibco Laboratories (Grand Island, NY, USA), respectively. The human hepatoma cell line HepG2 and the hepatic stellate cell line LX2 were obtained from American Type Culture Collection (Manassas, VA, USA).

#### Subjects

In the present investigation, a retrospective study was conducted on samples obtained from 203 obese subjects who were undergoing bariatric surgery. The detailed description of these subjects and the process of sample collection process were reported previously [45]. Collection of these samples was originally approved by the Institutional Review Board (IRB) of the Medical College of Wisconsin Froedtert Hospital (PRO ID: PRO00005335).

Based on their liver histology, subjects were classified into 80 apparently healthy controls (C), 93 patients with simple steatosis (SS), 16 patients with borderline NASH (B-NASH) and 14 patients with NASH [45]. NAFLD was diagnosed based on histological assessment after exclusion of other causes of liver disease. Either SS, B-NASH or NASH was diagnosed according to steatosis grade, steatosis distribution, microvesicular steatosis, ballooning, lobular inflammation, portal inflammation, fibrosis, NAFLD activity score (NAS) and elevated liver enzymes. The NAS score is the sum of scores for steatosis, lobular inflammation, and ballooning, and it ranges from 0 to 8 [26].

### 4.2. Methods

#### 4.2.1. Lipidomic Analysis

Targeted lipidomic analysis of lipids was conducted using an automated electrospray ionization (ESI)-tandem mass spectrometry approach according to a protocol described previously [46]. Data acquisition was performed based on a modified procedure by Devaiah et al. [47]. Briefly, serum lipids were extracted according to Folch method [48], then lipid extracts were dissolved in chloroform/methanol (9:1) prior to analysis. Precise amounts of internal standards for various phosphatidylcholines (PCs), lysophosphatidylcholines (LPCs), phosphatidylethanolamines (PEs), lysophosphatidylethanolamines (LPEs), phosphatidic acids (PAs), phytanoyl PAs, phosphatidylserines (PSs), phytanoyl PSs and phosphatidylinositols (PIs) were added. After mass spectrometry analysis, the data were normalized, and the background of each spectrum was deducted. Peaks for the target lipids in these spectra were then identified, followed by data correction for isotopic overlap, and molar amount calculation by comparing to the internal standards in the same lipid class. Data were expressed as the percentage of total lipids signal.

#### 4.2.2. Cell Culture

Hepatoma cells HepG2 and human hepatic stellate cells LX2 were maintained in high glucose Dulbecco’s modified eagle medium (DMEM) supplemented with 10% fetal bovine serum (FBS) and low glucose DMEM supplemented with 2% FBS, respectively, 100 U/mL penicillin G and 100 mg/mL streptomycin sulfate in a 5% CO_2_-humidified incubator at 37 °C. Both cells were periodically authenticated using microsatellite markers.

#### 4.2.3. Phospholipid Treatment

The cell study comprised three groups: a control group, a PE (34:1) group and a PE (36:2) group. Concentrated PE stocks were prepared in 2:1 (*v*/*v*) chloroform/methanol mixtures and stored in glass tubes under nitrogen at −20 °C. The desired amount was dried in glass tubes under a stream of nitrogen. Cell lines were treated with different concentrations of PEs [0.25 mM, 0.5 mM, and 1 mM of each of PE (34:1) and PE (36:2)]. The control group received the same amount of the vehicle. PEs were conjugated with 2% fatty acid-free bovine serum albumin.

#### 4.2.4. Cell Viability Assay

HepG2 cells were plated at an initial density of 5 × 10^3^ cells/well in a 96-well plate and treated with various concentrations (0.25, 0.5 and 1 mM) of PE (34:1), PE (36:2), or vehicle, for 24, 48 and 72 h. After incubation, 10 µL of Cell Counting Kit-8 (CCK8) (ApexBio, Houston, TX, USA) was added to each well of the 96-well microplate. The plate was placed in a CO_2_ incubator for 1–4 h to react. The absorbance was measured at 450 nm by a microplate reader.

#### 4.2.5. Lipid Droplet/Nucleus Staining with BODIPY/Hoechst

Lipid accumulation was assessed using fluorescent detection with BODIPY 493/503 (Invitrogen/Molecular Probes, Eugene, OR, USA) according to the method of Baumann et al. [49]. HepG2 cells were seeded in a 96-well plate at a density of 5 × 10^3^ cells/well, then the cells were incubated at 37 °C overnight. Thereafter, the cells were treated with vehicle or 1 mM PE (34:1 and 36:2) and incubated for 48 h. Fixation of cells was performed by adding 100 μL of 5% paraformaldehyde (PFA) solution in phosphate buffer saline (PBS) into the culture media to achieve a final PFA concentration of 2.5%, then incubation was allowed for 15 min at room temperature (RT). PFA was carefully removed, and the cells were washed twice with 100 μL PBS. BODIPY/Hoechst stock solutions were prepared to a working concentration of 5 μg/mL BODIPY and 1 μg/mL Hoechst, then 100 μL of both dyes were added in parallel and incubated for 15 min at RT. After incubation, the cells were washed twice with PBS, then 130 μL of PBS were added, and the resulting fluorescence intensity was measured at 493/503 for BODIPY and 352/454 for Hoechst by a microplate reader. To visualize the lipid droplets (LD) under the fluorescence microscope, the cells were cultured in a 35 mm culture dish overnight, then treated as above and thereafter stained with 5 μg/mL BODIPY and 1 μg/mL Hoechst. The LD were observed under the trinocular phase contrast fluorescence microscope (ZEISS Axiovert 200M, Carl Zeiss, Göttingen, Germany).

#### 4.2.6. Mitochondrial ROS Production/Nucleus Staining by MitoSox™/Hoechst

Mitochondrial reactive oxygen species (ROS) production was assessed by a red mitochondrial superoxide indicator MitoSox (Invitrogen/Molecular Probes, Eugene, OR, USA) according to the manufacturer’s instructions. HepG2 cells were cultured at a density of 5 × 10^3^ cells/well in a 96-well plate, then incubated at 37 °C overnight. The cells were incubated with vehicle or 1 mM PE (34:1 and 36:2) for 48 h. The culture medium was carefully removed and PBS containing 1 μg/mL Hoechst and 5 μM MitoSox was added to the cells, followed by incubation at 37 °C for 30 min. The staining solution was aspirated, the cells were washed twice with PBS, then the resulting fluorescence was measured using a plate reader at excitation/emission of 510/580 nm for MitoSox and 352/454 nm for Hoechst. For visualization of ROS production underneath the fluorescence microscope, the cells were cultured in a 35 mm culture dish overnight, treated as above, stained with 5 μM MitoSox and 1 μg/mL Hoechst, then observed under the trinocular phase contrast fluorescence microscope (ZEISS Axiovert 200M, Göttingen, Germany).

#### 4.2.7. Mitochondrial Membrane Potential (ΔΨm) and Mitochondrial Mass/Nucleus Staining

Mitochondrial membrane potential and mitochondrial mass were determined by staining with Mitotracker green and Mitotracker red (Invitrogen/Molecular Probes, Eugene, OR, USA), respectively, according to the manufacturer’s instructions. HepG2 cells were cultured at a density of 5 × 10^3^ cells/well in a 96-well plate overnight, treated with vehicle or 1 mM PE (PE 34:1 and PE 36:2) for 48 h. Mitochondrial mass and mitochondrial membrane potential were determined by staining cells with 100 nM, 200 nM, and 1 μg/mL Mitotracker red, green and Hoechst, respectively. Fluorescence intensity of Mitotracker red, green and Hoechst was assessed by a plate reader at excitation/emission of 579/599 nm, 490/516 nm and 352/454, respectively. To examine alteration in mitochondrial mass and membrane potential under the trinocular phase contrast fluorescence microscope (ZEISS Axiovert 200M, Göttingen, Germany), the cells were plated in a 35 mm culture dish overnight, treated as above, then stained.

#### 4.2.8. Cell Migration Assay

Cell migration was assayed with a Transwell plate according to the method described by Justus et al. [50]. In a 24-well Transwell with 8.0 µm pore polycarbonate membrane inserts (CORNING, Corning, NY, USA), 2 × 10^4^ LX2 cells were plated in serum-free low glucose DMEM with vehicle, 1 mM PE (34:1 and 36:2), or 0.04 μg/mL lipopolysaccharide (LPS) as a positive control. An aliquot of 500 μL of low glucose DMEM containing 10% FBS as chemoattractant was added to the lower chamber. The cells were incubated at 37 °C and 5% CO_2_ for 24 h. Migrated cells were fixed with 70% ethanol, stained with 0.2% crystal violet and pictures were captured underneath an upright digital microscope (SeBa™–Laxco, Mill Creek, WA, USA). Crystal violet was extracted by 33% acetic acid, transferred to a 96-well clear microplate, and the absorbance at 595 nm was measured using a plate reader.

#### 4.2.9. RNA Extraction and Reverse Transcription Polymerase Chain Reaction (RT-PCR)

RNA was extracted using TRIzol^®^ Reagent (Life Technologies Corporation, Carlsbad, CA, USA) from HepG2 and LX2 cell suspensions pretreated with vehicle or 1 mM PE (34:1 and 36:2) for 48 h prior to extraction according to the manufacturer’s instructions. Concentration and purity of RNA were measured spectrophotometrically at A280 and A260. For quantification of mRNA expression levels, RNA was reversely transcribed using reverse transcription cDNA Synthesis kit (ThermoFisher Scientific, Waltham, MA, USA). The cDNA was used as template in qPCR Applied Biosystems QuantStudio™ 7 Flex Real-Time PCR System using Universal SYBR Green (BioRad, Irvine, CA, USA) and gene-specific primer pairs listed in Appendix A.

The mRNA expression levels of carnitine palmitoyltransferase (*CPT*), peroxisome proliferator-activated receptor-alpha (*PPARa*), diglyceride acyltransferase (*DGAT*), fatty acid synthase (*FASN*), sterol regulatory element-binding protein (*SREBP*), mechanistic target of rapamycin kinase (*mTOR*), farnesoid X receptor gene (FXR), carbohydrate response element binding protein (*ChREBP*), microtubule-associated protein (*LC3*), Bcl-2 Associated X-protein (*BAX*), p21-activated kinase 2 (*PAK2*), cytochrome C (*CYCs*), B-cell lymphoma 2 (*BCL2*), alpha-smooth muscle actin (*α-SMA*), collagen type I alpha 1 (*COL1A1*), collagen type III alpha 1 (*COL3A1*), TIMP metallopeptidase inhibitor 1 (*TIMP1*), transforming growth factor beta-1 (*TGFbeta*), tumor necrosis factor-alpha (*TNFα*), TIMP metallopeptidase inhibitor 3 (*TIMP3*) and interleukin-6 (*IL6*) were determined by quantitative real-time polymerase chain reaction (qRT-PCR). The relative levels of mRNA expression were calculated after normalizing to the endogenous control glyceraldehyde-3-phosphate dehydrogenase (*GAPDH*). Fold changes were calculated for relative quantification (2DDCt).

#### 4.2.10. Statistics

Statistical analysis was performed by the statistical software R and GraphPad Prism 7.4 software. Continuous data were checked for normality by Shapiro-Wilk’s test. For patient and clinical characteristics, count and percentage were used to summarize categorical variables and, for continuous variables, median and range were used. Other continuous data (such as % of total lipids signal, cell viability, staining, fluorescence intensity, mRNA expression, etc.) were expressed as mean ± standard deviation (SD). Parametric data were analyzed by unpaired t-test, one-way ANOVA, or two-way ANOVA. Unpaired t-test was performed to compare between two groups. One-way ANOVA was carried out to compare among three or more groups, followed by Tukey’s or Holm’s post-hoc pairwise comparisons. Correspondingly, two-way ANOVA was used to analyze differences between three or more independent groups that had been split on two variables (time-dependent data), followed by Tukey’s or Holm’s post-hoc analysis. Non-parametric data were analyzed using Kruskal-Wallis test, followed by Dunn’s or Holm’s post-hoc pairwise comparisons. Categorical data were analyzed using Fisher’s exact test, and *p* values less than 0.05 were considered statistically significant.

## Figures and Tables

**Figure 1 ijms-24-01034-f001:**
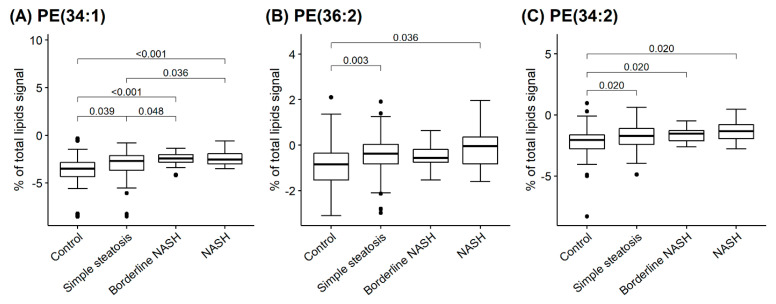
Circulating phospholipids in control (n = 80), SS (n = 93), B-NASH (n = 16) and NASH (n = 14) subjects. Box plots of lipid profiling of 203 subjects generated by ESI-MS/MS method. Quantification of (**A**) Phosphatidylethanolamine 34:1 (PE 34:1); (**B**) Phosphatidylethanolamine 36:2 (PE 36:2); (**C**) Phosphatidylethanolamine 34:2 (PE 34:2); (**D**) Cholesterol ester (C16:1 CE); (**E**) ratio of lysophosphatidylcholine 18:2/phosphatidylethanolamine 34:2 (LPC 18:2/PE 34:2); (**F**) ratio of lysophosphatidylcholine 18:2/phosphatidylethanolamine 34:1 (LPC 18:2/PE 34:1); (**G**) ratio of lysophosphatidylcholine 18:2/phosphatidylethanolamine 36:2 (LPC 18:2/PE 36:2); (**H**) ratio of lysophosphatidylcholine 18:2/cholesterol ester C16:1 (LPC 18:2/C16:1 CE); and (**I**) ratio of cholesterol ester C16:1/cholesterol ester C18:1 (C16:1 CE/C18:1 CE) in the serum of control, simple steatosis (SS), borderline non-alcoholic steatohepatitis (BNASH) and non-alcoholic steatohepatitis (NASH) patients in serum. *p*-values were obtained from post-hoc pairwise comparisons after one-way ANOVA.

**Figure 2 ijms-24-01034-f002:**
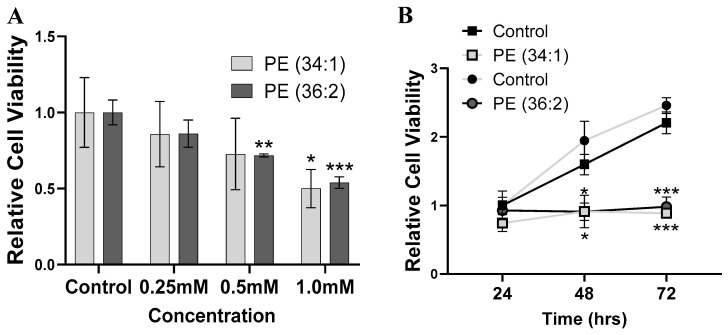
Dose and time dependent effects of phosphatidylethanolamine (PE) on the viability of HepG2 cells. CCK8 assay was performed after treatment with (**A**) different doses (0.25 mM, 0.5 mM, and 1 mM) of PE (34:1) and PE (36:2) for 48 h. (**B**) Single dose (1 mM) of PE (34:1) and PE (36:2) for 24, 48 and 72 h. Results are depicted as relative change of control. Bars represent the means ± SDs of three independent experiments * *p* < 0.05, ** *p* < 0.01, and *** *p* < 0.001 vs. control.

**Figure 3 ijms-24-01034-f003:**
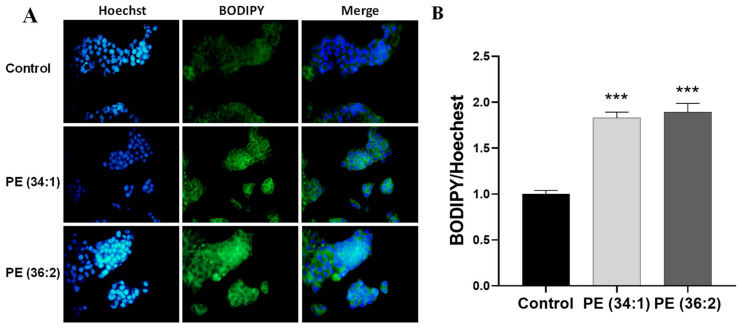
Lipid accumulation in HepG2 cells detected by BODIPY/Hoechst staining. Cells were treated with 1 mM phosphatidylethanolamine (PE) for 48 h. (**A**) Representative photomicrographs of HepG2 cells (×40). (**B**) Quantification of fluorescence intensity of intracellular fat drops by plate reader at excitation/emission of 493/503 nm. Results are depicted as relative change of control. Bars represent the means ± SDs of three independent experiments *** *p* < 0.001 vs. control.

**Figure 4 ijms-24-01034-f004:**
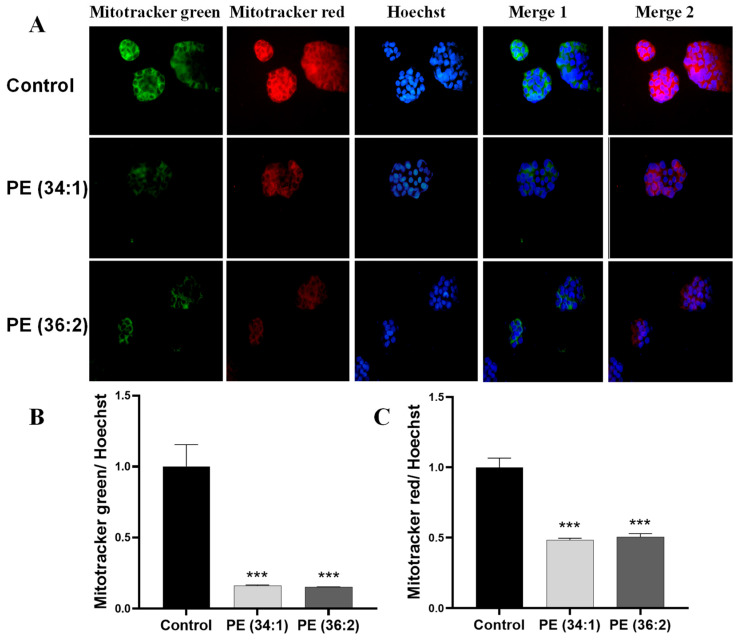
Mitochondrial dysfunctions indexed by the reduction in mitochondrial mass and membrane potential assessed by using Mitotracker green and red staining, respectively. Cells were treated with 1 mM phosphatidylethanolamine (PE) for 48 h. (**A**) Representative photomicrographs of HepG2 cells (×40). (**B**) Quantification of fluorescence intensity of mitochondrial mass by plate reader at excitation/emission of 490/516 nm for Mitotracker green. (**C**) Quantification of fluorescence intensity of mitochondrial membrane potential by plate reader at excitation/emission of 579/599 nm for Mitotracker red. Results are depicted as the ratio of Mitotracker/Hoechst relative to control. Bars represent the means ± SDs of three independent experiments, *** *p* < 0.001 vs. control.

**Figure 5 ijms-24-01034-f005:**
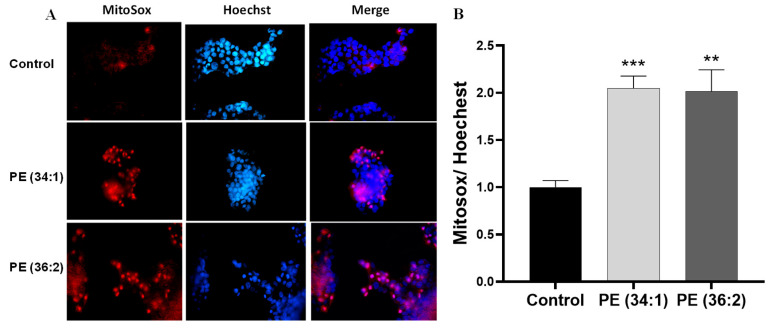
Mitochondrial ROS production/nucleus in HepG2 cells detected by MitoSox/Hoechst staining. Cells were treated with 1 mM phosphatidylethanolamine (PE) for 48 h. (**A**) Representative photomicrographs of HepG2 cells (×40). (**B**) Quantification of fluorescence intensity of ROS generation by plate reader at excitation/emission of 510/580 nm. Results are depicted as relative change of control. Bars represent the means ± SDs of three independent experiments. ** *p* < 0.01, and *** *p* < 0.001 vs. control.

**Figure 6 ijms-24-01034-f006:**
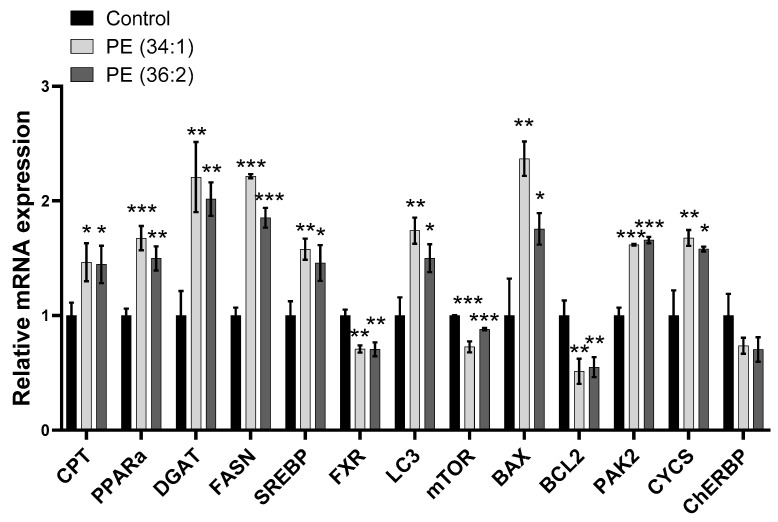
Gene expression alteration in HepG2 cells. Treatment of cells with 1 mM phosphatidylethanolamine (PE) for 48 h induced changes in the expression of genes involved in ꞵ-oxidation (*CPT* and *PPARa*), lipogenesis (*DGAT*, *FASN*, *SREBP* and *FXR*), mitophagy (*LC3* and *mTOR*) and apoptosis (*BAX*, *BCL2*, *PAK2*, *ChERBP*, and *CYCS*). Results are depicted as relative change of control. Data represent the mean ± SD of three independent experiments. * *p* < 0.05, ** *p* < 0.01 and *** *p* < 0.001 vs. control. *CPT*, carnitine palmitoyltransferase; *PPARa*, peroxisome proliferator-activated receptor-alpha; *DGAT*, diglyceride acyltransferase; *FASN*, fatty acid synthase; *SREBP*, sterol regulatory element-binding protein; *FXR*, farnesoid X receptor gene; *LC3*, microtubule-associated protein; *mTOR*, mechanistic target of rapamycin kinase; *BAX*, Bcl-2 associated X-protein; *BCL2*, B-cell lymphoma 2; *PAK2*, p21-activated kinase 2; *ChREBP*, carbohydrate response element binding protein; *CYCs*, cytochrome C.

**Figure 7 ijms-24-01034-f007:**
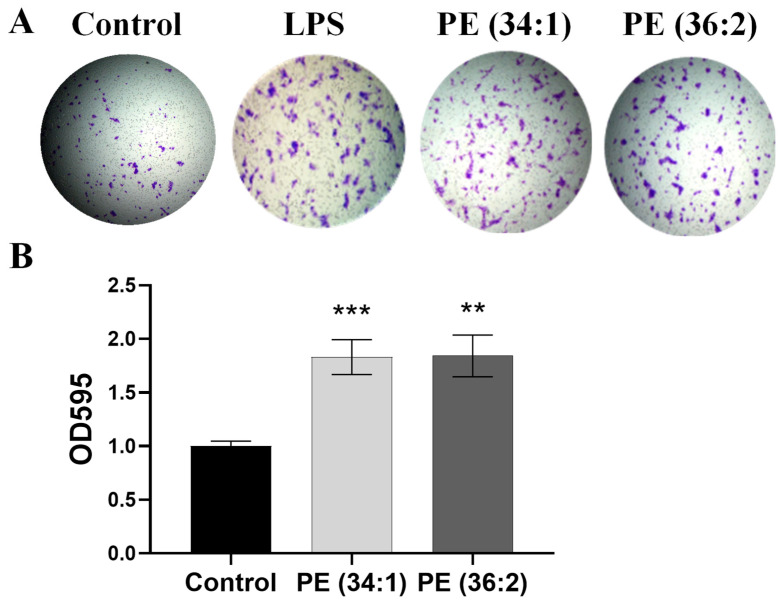
Phosphatidylethanolamine (PE) promotes LX2 cell migration. (**A**) Representative image of the bottom surface of a Transwell migration assay after incubation with 1 mM PE or lipopolysaccharide (LPS) (0.04 μg/mL) as a positive control for 24 h. (**B**) The absorbance values of migratory cells. Results are depicted as relative change of control at OD595. Bars represent the means ± SDs of three independent experiments. ** *p* < 0.01, and *** *p* < 0.001 vs. control.

**Figure 8 ijms-24-01034-f008:**
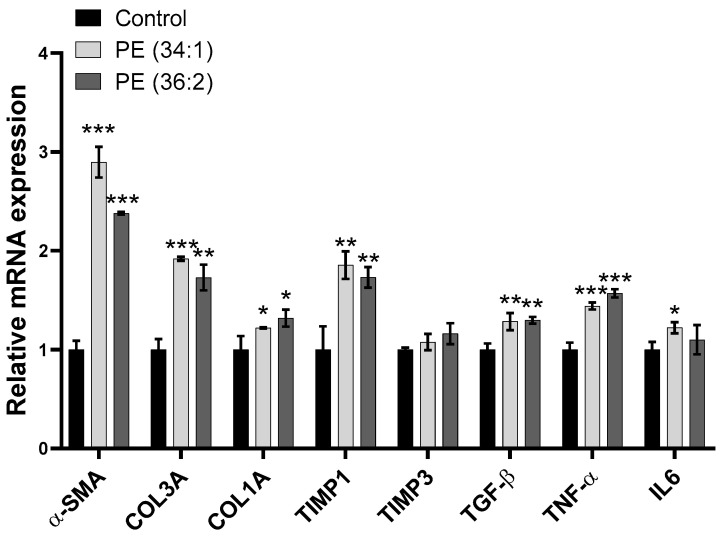
Gene expression alteration in LX2 cells. Treatment of cells with 1 mM phosphatidylethanolamine (PE) for 48 h induced changes in the expression of genes involved in fibrosis (*α-SMA*, *COL1A1, COL3A1, TIMP1, TIMP3 and TGFꞵ*) and inflammation (*TNFα* and *IL6*). Results are depicted as relative change of control. Data represent the mean ± SD of three independent experiments. * *p* < 0.05, ** *p* < 0.01, and *** *p* < 0.001 vs. control. *α-SMA*, alpha-smooth muscle actin; *COL1A1*, collagen type I alpha 1, *COL3A1*, collagen type III alpha 1; *TIMP1*, TIMP metallopeptidase inhibitor 1; *TIMP3*, TIMP metallopeptidase inhibitor 3; *TGFꞵ*, transforming growth factor beta-1; *TNFα*, tumor necrosis factor-alpha; and *IL6*, interleukin-6.

**Table 1 ijms-24-01034-t001:** Demographic characteristics of study subjects.

	Control (n = 80)	Simple Steatosis (n = 93)	Borderline NASH (n = 16)	NASH (n = 14)	All (n = 203)	*p* *
Age (years)–Median (range)	42 (20, 66)	46 (20, 66)	43.5 (32, 58)	49.5 (26, 58)	44 (20, 66)	0.348
Gender–no. (%)						0.091
Female	73 (91)	86 (92)	14 (88)	10 (71)	183 (90)	
Male	7 (9)	7 (8)	2 (12)	4 (29)	20 (10)	
Race–no. (%)						0.545
EA	69 (86)	77 (83)	12 (75)	13 (93)	171 (84)	
Non-EA	11 (14)	16 (17)	4 (25)	1 (7)	32 (16)	
Height (feet)–Median (range)	5.5 (4.92, 6.17)	5.42 (4.92, 6.25)	5.42 (5, 5.67)	5.54 (4.75, 6.25)	5.42 (4.75, 6.25)	0.248
Weight (lbs)–Median (range)	278.5 (194, 481)	291 (194, 381)	288 (207, 502)	310.5 (217, 416)	281.5 (194, 502)	0.768
Diabetes Mellitus–no. (%) ^‡^						<0.001
No	68 (85)	61 (66)	8 (50)	5 (36)	142 (70)	
Yes	12 (15)	32 (34) ^‡^	8 (50) ^‡^	9 (64) ^‡^	61 (30)	
Hypertension–no. (%)						0.148
No	43 (54)	45 (48)	9 (56)	3 (21)	100 (49)	
Yes	37 (46)	48 (52)	7 (44)	11 (79)	103 (51)	
Dyslipidemia–no. (%)						0.587
No	59 (74)	68 (73)	10 (62)	12 (86)	149 (73)	
Yes	21 (26)	25 (27)	6 (38)	2 (14)	54 (27)	
Depression–no. (%)						0.673
No	45 (56)	59 (63)	10 (62)	7 (50)	121 (60)	
Yes	35 (44)	34 (37)	6 (38)	7 (50)	82 (40)	

* *p* value is calculated by the Kruskal-Wallis test for continuous variables and Fisher’s exact test for categorical variables among four groups. ‡ Post-hoc *p* values for pairwise comparisons with control are 0.008, 0.008 and <0.001 for simple steatosis, borderline NASH, and NASH, respectively. NASH, nonalcoholic steatohepatitis; EA, European; non-EA, non-European; DM, diabetes mellitus.

**Table 2 ijms-24-01034-t002:** Histological features of study subjects.

	Control (n = 80)	Simple Steatosis (n = 93)	Borderline NASH (n = 16)	NASH (n = 14)	All (n = 203)	*p* *
Steatosis grade–no. (%)						<0.001
<5%	80 (100)	0 (0)	0 (0)	1 (7)	81 (40)	
5–33%	0 (0)	58 (62)	9 (56)	4 (29)	71 (35)	
33–66%	0 (0)	24 (26)	6 (38)	6 (43)	36 (18)	
>66%	0 (0)	11 (12)	1 (6)	3 (21)	15 (7)	
Steatosis distribution–no. (%)						<0.001
Zone 3	76 (95)	32 (34)	3 (19)	6 (43)	117 (58)	
Zone 1	0 (0)	2 (2)	1 (6)	1 (7)	4 (2)	
Azonal	2 (2)	39 (42)	10 (62)	3 (21)	54 (27)	
Panacinar	0 (0)	20 (22)	2 (12)	4 (29)	26 (13)	
Missing	2 (2)	0 (0)	0 (0)	0 (0)	2 (1)	
Microvesicular steatosis–no. (%)						<0.001
Not present	78 (98)	22 (24)	1 (6)	1 (7)	102 (50)	
Present	2 (2)	70 (75)	15 (94)	13 (93)	100 (49)	
Missing	0 (0)	1 (1)	0 (0)	0 (0)	1 (0)	
Ballooning–no. (%)						<0.001
None	80 (100)	93 (100)	10 (62)	5 (36)	188 (93)	
Few balloon cells	0 (0)	0 (0)	6 (38)	8 (57)	14 (7)	
Many cells/prominent ballooning	0 (0)	0 (0)	0 (0)	1 (7)	1 (0)	
Lobular inflammation–no. (%)						<0.001
No foci	80 (100)	93 (100)	13 (81)	9 (64)	195 (96)	
2 foci per 200 field	0 (0)	0 (0)	3 (19)	5 (36)	8 (4)	
Portal inflammation–no. (%)						<0.001
None	74 (92)	68 (73)	6 (38)	2 (14)	150 (74)	
Mild	6 (8)	24 (26)	9 (56)	10 (71)	49 (24)	
Moderate	0 (0)	1 (1)	1 (6)	2 (14)	4 (2)	
Fibrosis–no. (%)						<0.001
None	80 (100)	91 (98)	8 (50)	5 (36)	184 (91)	
Perisinusoidal or periportal	0 (0)	2 (2)	4 (25)	6 (43)	12 (6)	
Perisinusoidal and portal/periportal	0 (0)	0 (0)	1 (6)	0 (0)	1 (0)	
Cirrhosis	0 (0)	0 (0)	3 (19)	3 (21)	6 (3)	
Mallory bodies–no. (%)						<0.001
None to rare	80 (100)	93 (100)	15 (94)	11 (79)	199 (98)	
Many	0 (0)	0 (0)	1 (6)	3 (21)	4 (2)	
NAS score–median (range) ^‡^	0 (0, 0)	1 (1, 3)	2 (1, 3)	3 (0, 5)	1 (0, 5)	<0.001

* *p* value is calculated by the Kruskal-Wallis test for continuous variables and Fisher’s exact test for categorical variables among four groups. ‡, NAFLD activity scores (NAS) were generated according to the NASH clinical research network scoring system proposed by Kleiner et al. [26].

**Table 3 ijms-24-01034-t003:** Biochemical parameters of study subjects.

	Median (Range)					*p* *
	Control (n = 80)	Simple Steatosis (n = 93)	Borderline NASH (n = 16)	NASH (n = 14)	All (n = 203)	
T Chol	169.5 (104, 278)	178.5 (112, 268)	174 (107, 295)	171 (117, 266)	173 (104, 295)	0.702
Missing	0	3	0	1	4	
TG	107 (41, 600)	147.5 (49, 693)	144.5 (63, 968)	218 (66, 823)	126 (41, 968)	<0.001
Missing	0	3	0	1	4	
HDL	45.5 (30, 81)	41 (24, 69)	38.5 (26, 64)	40 (29, 61)	43 (24, 81)	0.001
Missing	0	3	0	1	4	
LDL	99.5 (38, 163)	103 (46, 188)	108 (49, 138)	91.5 (43, 154)	103 (38, 188)	0.895
Missing	2	6	2	4	14	
Albumin	4.2 (3.8, 64)	4.2 (0.1, 73)	4.2 (3.5, 5.1)	4.25 (3.8, 4.5)	4.2 (0.1, 73)	0.968
Missing	1	2	0	0	3	
HbA1c	6.3 (5.7, 10.1)	6.5 (3.1, 13.9)	6.8 (5.5, 11.4)	6.5 (5.3, 9.7)	6.5 (3.1, 13.9)	0.612
Missing	70	62	6	7	145	
Insulin level	20.43 (7.04, 44.58)	25.56 (7.14, 112)	24.13 (11.76, 178.9)	32.88 (11.47, 52.69)	24.09 (7.04, 178.9)	0.058
Missing	44	39	7	5	95	
ALT	17 (8, 74)	22 (9, 65)	40.5 (10, 99)	30 (7, 85)	20 (7, 99)	<0.001
Missing	1	2	0	0	3	
AST	16 (7, 55)	19 (11, 76)	35.5 (15, 127)	29.5 (15, 55)	19 (7, 127)	<0.001
Missing	1	2	0	0	3	

* *p* value is calculated by the Kruskal-Wallis test among four groups. NASH, nonalcoholic steatohepatitis; T Chol, total cholesterol; TG, triglycerides; HDL, high density lipoprotein; LDL, low density lipoprotein; HbA1c, Hemoglobin A1C; ALT, alanine aminotransferase; AST, aspartate aminotransferase.

## Data Availability

The authors declare that the data generated and analyzed during this study are included in this published article and associated Appendix A. In addition, datasets generated and/or analyzed during the current study are available from the corresponding author on reasonable request.

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
