# Peer review of "Phosphatidylethanolamines Are Associated with Nonalcoholic Fatty Liver Disease (NAFLD) in Obese Adults and Induce Liver Cell Metabolic Perturbations and Hepatic Stellate Cell Activation"

_ijms, 2023, doi:10.3390/ijms24021034_

Round 1
Reviewer 1 Report
Authors have rationally designed the project and seems like all the experiments were carried out meticulously. I have thoroughly enjoyed reading the results. I do hereby recommend this manuscript.
1. In the manuscript title, “Are” should not be capitalized.
Here are the specific comments for this accepted manuscript. Query: What is the main question addressed by the research? Is it relevant and interesting? Response: In this research, authors have established the relationship between phospholipids, especially phosphatidylethanolamines with Nonalcoholic fatty liver disease with a focus on obese patients. This study was quite interesting as it covered a sufficient number of obese patients (more than 200). Further, a detailed role was elucidated with the help of in vitro- cell line-based studies.Query: How original is the topic? What does it add to the subject area compared with other published material? Response: The topic was quite original. Patients covered by the authors are covering patients with simple hepatic steatosis (SS), borderline non-alcoholic steatohepatitis (B-NASH) and progressive NASH (NASH). A strong association of phosphatidylethanolamines (PEs), PE(34:1) and PE(36:2) with NAFLDs was established in this study. When I did the search through other published materials, I noticed that this study seems to be the first of its kind where PEs are associated with NAFLD in obese patients.
Query: Is the paper well written? Is the text clear and easy to read? Response: Yes, the paper is very well written. Text is also very clear and easy to read.
Query: Are the conclusions consistent with the evidence and arguments presented? Do they address the main question posed? Response: Yes, conclusions are consistent with the evidence and arguments presented. And also, they have addressed the main questions. It is very rare that I endorse any manuscript on the first attempt. But I found this paper very interesting and worthy for readers. That is why I straightforwardly recommended it.
Author Response
Authors have rationally designed the project and seems like all the experiments were carried out meticulously. I have thoroughly enjoyed reading the results. I do hereby recommend this manuscript.
Response: We thank the reviewer’s careful review and appreciation for our manuscript.
- In the manuscript title, “Are” should not be capitalized.
Response: We now revised this accordingly.

Reviewer 2 Report
Can the author include a western blot for gene expression presented in the figure 6 & 8.
Author Response
Can the author include a western blot for gene expression presented in the figure 6 & 8.
Response: We thank this reviewer’s suggestion. It is indeed a limitation that our work did not include WB, which should provide additional information for supporting our work. However, we also politely deem that qPCR data has sufficiently supported our conclusion. These genes are well-established markers for characterizing liver cell injury and HSC activation. In the field of NAFLD research, the transcription levels of these marker genes are broadly measured by either RNA-seq or qPCR to verify molecular changes underlying perturbations in liver cells during NASH development. We have mentioned this limitation in our discussion session.

Reviewer 3 Report
Nonalcoholic fatty liver disease (NAFLD), a chronic liver disease, is frequently observed in elderly people and in patients with obesity, type 2 diabetes mellitus, hypertension, and metabolic syndrome. Despite obesity has been considered a prerequisite factor of NAFLD, NAFLD is also observed in non-obese individuals. In this study, the disruption of phospholipids (PE) in the liver might be associated with the progression of NAFLD in obese individuals.
No serious problems were found with the methods, data analysis, or discussion, and the manuscript was evaluated as worthy of publication in this journal.
The following are some of the points of interest in this manuscript
1. The enrolled subjects are all obese. I understand that this study tried to identify the association of PEs in patients with NAFLD related to obesity. However, it would have been better if the PE level of non-obese NAFLD patients was included.
2. In order to identify the role of PEs as an obesity-related NAFLD pathogenesis, it seems necessary to observe changes in fibrosis or inflammation-related indicators when treated with substances that inhibit the action of PEs.
Author Response
Comments and Suggestions for Authors
Nonalcoholic fatty liver disease (NAFLD), a chronic liver disease, is frequently observed in elderly people and in patients with obesity, type 2 diabetes mellitus, hypertension, and metabolic syndrome. Despite obesity has been considered a prerequisite factor of NAFLD, NAFLD is also observed in non-obese individuals. In this study, the disruption of phospholipids (PE) in the liver might be associated with the progression of NAFLD in obese individuals.
No serious problems were found with the methods, data analysis, or discussion, and the manuscript was evaluated as worthy of publication in this journal.
Response: We thank this reviewer’s careful review and positive comments.
The following are some of the points of interest in this manuscript
- The enrolled subjects are all obese. I understand that this study tried to identify the association of PEs in patients with NAFLD related to obesity. However, it would have been better if the PE level of non-obese NAFLD patients was included.
Response: We agree with this comment. However, as we have mentioned in our manuscript, a few published studies in the area have investigated the relationship between phospholipids, especially PCs and PEs and NAFLD in non-obese patients (references 20 & 36). Our work is designed to specifically study the relationship between PLs and obese NAFLD, which is a knowledge gap in the field.
- In order to identify the role of PEs as an obesity-related NAFLD pathogenesis, it seems necessary to observe changes in fibrosis or inflammation-related indicators when treated with substances that inhibit the action of PEs.
Response: We completely agree with this comment. Unfortunately, the homeostasis, transportation, biosynthesis and metabolism of PEs are very complicated, which may not be readily probed by a single inhibitor. More importantly, the key mediators of the PE signaling remain largely unknown. We are now conducting additional works to identify critical molecules mediating the impact of PEs. The work presented in our manuscript provided proof-of-concept evidence that PEs are indeed causally involved in liver cell injury and HSC activation, indicating the existence of such mediators. We have now included this limitation in the discussion.
